# Management of veterinary anaesthesia and analgesia in small animals: A survey of English-speaking practitioners in Canada

Sophie Lalonde[1], Geoffrey Truchetti[1,2], Colombe Otis[3], Guy Beauchamp[3], Eric Troncy[3]*

1 Centre Vétérinaire Rive-Sud, Brossard, Québec, Canada, 2 Centre Vétérinaire Laval, Laval, Québec, Canada, 3 Faculty of Veterinary Medicine, Groupe de Recherche en Pharmacologie Animale du Québec (GREPAQ), Université de Montréal, Saint-Hyacinthe, Québec, Canada

* eric.troncy@umontreal.ca

## Abstract

### Objective

To describe how small animal anaesthesia and analgesia is performed in English-speaking Canada, document any variation among practices especially in relation to practice type and veterinarian's experience and compare results to published guidelines.

### Design

Observational study, electronic survey.

### Sample

126 respondents.

### Procedure

A questionnaire was designed to assess current small animal anaesthesia and analgesia practices in English-speaking Canadian provinces, mainly in Ontario, Alberta and British Columbia. The questionnaire was available through SurveyMonkey® and included four parts: demographic information about the veterinarians surveyed, evaluation and management of anaesthetic risk, anaesthesia procedure, monitoring and safety. Year of graduation and type of practice were evaluated as potential risk factors. Exact chi-square tests were used to study the association between risk factors and the association between risk factors and survey responses. For ordinal data, the Mantel-Haenszel test was used instead.

### Results

Response rate over a period of 3 months was 12.4% (126 respondents out of 1 016 invitations). Current anaesthesia and analgesia management failed to meet international guidelines for a sizable number of participants, notably regarding patient evaluation and preparation, safety and monitoring. Nearly one third of the participants still consider

**Data Availability Statement:** All relevant data are within the paper and its Supporting information files.

**Funding:** There was not proprietary interest or funding directly provided for this project. This work was indirectly supported (ETR) by a Discovery grant (#441651–2013, supporting salaries) and a Collaborative Research and Development grant (#RDCPJ 491953–2016 supporting operations and salaries in partnership with ArthroLab Inc.) from the Natural Sciences and Engineering Research Council of Canada. COT is a recipient of a MITACS Canada Elevation postdoctoral scholarship (#IT11643). The authors got support from the company Dispomed Inc., i.e. to deliver the electronic survey to their clients. The funders had no role in study design, data collection and analysis, decision to publish, or preparation of the manuscript.

**Competing interests:** The authors have read the journal's policy and the authors of this paper have the following competing interests: ArthroLab Inc., as partner in a funding grant. The joint ArthroLab – NSERC funding only participated indirectly by covering partially the salary of COT. The specific roles of this author are articulated in the 'author contributions' section. The authors received support from the company Dispomed Inc., i.e. to deliver the electronic survey to their clients. This has been detailed in the text. The funders, either governmental (NSERC and MITACS) or commercial (ArthroLab Inc., Dispomed Inc.) had no role in the present study. This does not alter our adherence to PLOS ONE policies on sharing data and materials. There are no patents, products in development or marketed products to declare.

analgesia as optional for routine surgeries. Referral centres tend to follow guidelines more accurately and are better equipped than general practices.

## Conclusions and clinical relevance

A proportion of surveyed Canadian English-speaking general practitioners do not follow current small animal anaesthesia and analgesia guidelines, but practitioners working in referral centres are closer to meet these recommendations.

## Introduction

Anaesthesia takes place almost every day in small animal veterinary practice. Several guidelines have been published, including recommendations for best practice in pre-anaesthetic work-up, anaesthetic monitoring, and analgesia [1–5]. A recently published survey described current French-speaking Eastern Canada veterinary anaesthesia management [6]. The authors concluded that the level of care in French-speaking Eastern Canada failed to meet published guidelines for several criteria. This was the most concerning finding, especially regarding analgesia standard of care (client prompted optional analgesia for 29% of respondents) [6]. Furthermore, they found that several demographic factors such as the type of veterinary practice, either general practice (GP) or referral centre, the veterinarian's gender and year of graduation influenced different aspects of anaesthesia [6]. It is currently unknown if this is unique to French-speaking veterinarians or if this situation is widespread across Canada or elsewhere in the world. The literature is rather scarce regarding what is actually done in clinical anaesthesia settings. Studies reported a wide range of anaesthetic and analgesic protocols within New Zealand [7], and between New Zealand and United Kingdom and Australia [8] for gonadectomy in dogs and cats, supporting that geographic localisation affects anaesthesia and analgesia practice. Obtaining a realistic portrait of current anaesthetic practice is essential to assess strengths and weaknesses and to improve standard of care by adapting veterinary cursus/curriculum and continuing education for recently graduated and future veterinarians. The objectives of this study were to describe the standards of small animal anaesthesia and analgesia by English-speaking veterinarians practicing in Canada, to compare them to published guidelines, and to examine which demographic factors influence anaesthesia management. Our hypothesis was that there would be discrepancies between the studied population standards and current guidelines. Furthermore, we hypothesised that the type of veterinary practice and the veterinarian's experience would influence anaesthetic care.

## Materials and methods

### Questionnaire

Members of the Research Group in Animal Pharmacology of Quebec (GREPAQ) developed a questionnaire (for detailed questions and choice of answers, see S1 Appendix), designed to assess current small animal anaesthesia and analgesia practices in English-speaking Canada. The internal content and construct validation included a pilot survey with a focus group. The latter included various degrees of expertise in veterinary anaesthesia, from veterinary student, general practitioner to anaesthesiologist in private practice and academia. They evaluated and validated all sections as well as all used terminology to be perfectly understood for any registered veterinary general practitioner, which was the expected audience of the survey. The

Ethics Committee for Research in Health and Sciences (CERSES) of Université de Montréal confirmed that such quality improvement in veterinary practice study fell under the Article 2.5 of the Tri-Council Policy Statement of Canada; Ethical Conduct of Research Involving Humans, 2nd edition 2014 (http://www.pre.ethics.gc.ca/eng/policy-politique/initiatives/tcps2-eptc2/Default/) of the activities not requiring research ethics board review.

The questionnaire was available through SurveyMonkey® *via* an electronic link that was sent by email by a veterinary equipment company (Dispomed Ltd.) to all their small animal veterinary customers. The survey consisted of four parts: Part I collected demographic information about the veterinarians surveyed. Part II focused on the evaluation and management of small animal anaesthetic risk. Part III investigated the anaesthesia procedure and finally, in Part IV, respondents evaluated the monitoring and safety of anaesthesia, including during the post-anaesthetic period. Response rate over a period of 3 months, March to May 2016, was 12.4% (126 respondents out of 1 016 invitations) amongst Canadian English-speaking small animal practitioners: invitations sent mostly in Ontario (n = 488), Alberta (n = 292), and British-Columbia (n = 148), as well as Manitoba, Saskatchewan, Maritimes, and Newfoundland.

## Statistical analysis

Two independent observers (COT, SLA) validated the data by first manually double-checking records from the SurveyMonkey® report, and then editing the descriptive statistics. For inferential statistical analysis, the selected demographic characteristics described in Part I, namely year of graduation and type of practice, were tested as potential risk factors influencing responses in the following sections. These factors were chosen based on the results of a previous study [6]. Indeed, they were likely to affect the results in the current study as well. Exact chi-square tests were used to examine the association between risk factors and the association between risk factors and survey responses. For ordinal data, the Mantel-Haenszel test was used instead. For descriptive purposes, we rely on percentages based on the number of responses because not all respondents answered all questions. Statistical analyses were performed with SAS v.9.4 (SAS Institute, Cary, NC, USA). Results are showed in percentage of the significant risk factor direction effect for each answer, and statistical *P*-value associated for the statistically significant difference (*P*-value ≤ 0.05).

## Results

### Part I—Demographic data

A total of seven demographic characteristics are presented in Table 1 with the distribution of each risk factor. Most veterinarians responding to the survey worked as general practitioners (GPs) in small practices (less than 5 veterinarians) either in very large city or small town, were not often on call, and anaesthetised only 2–3 animals per day.

**Risk factors.**   Significant associations occurred between risk factors and are summarised in Table 2. The type of practice and years of experience were tested for their potential influence on subsequent responses. To avoid redundant influence, number of animals anaesthetised per day (as it was associated to type of practice) and gender (as it was associated to year of graduation) were not considered further. Only demographic characteristics with a statistically significant influence are detailed below.

### Part II—Evaluation and management of anaesthetic risk

**Client management.**   Among respondents, 65% (82/126) provide handouts or other supporting material explaining anaesthesia procedure and related risk. GPs are more likely to

**Table 1. Demographic characteristics of English-speaking veterinarians (n = 126) responding to a survey on management of anaesthesia in small animal practices in Canada.**

| Characteristic | Distribution |
|---|---|
| Gender | |
| Male | 48/126 (38.1%) |
| Female | 78/126 (61.9%) |
| Years of practice since veterinary school graduation | |
| <15 years | 58/126 (46.0%) |
| >15 years | 68/126 (54.0%) |
| Number of veterinarian(s) in the practice | |
| 1 | 30/126 (23.8%) |
| 2–4 | 73/126 (57.9%) |
| 5+ | 23/126 (18.3%) |
| On-call duty[a] | |
| Yes[b] | 31/125 (24.8%) |
| Never | 70/125 (56.0%) |
| Other[c] | 24/125 (19.2%) |
| Size of town (population) | |
| Very large city (>100 000) | 45/126 (35.7%) |
| Large city (50 000 to 100 000) | 23/126 (18.3%) |
| Middle-size town (10 000 to 50 000) | 23/126 (18.3%) |
| Small town (<10 000) | 35/126 (27.8%) |
| Type of practice | |
| General practice (GP) | 113/126 (89.7%) |
| Referral centre | 13/126 (10.3%) |
| Number of animal(s) anaesthetised/day | |
| 0–1 | 23/126 (18.3%) |
| 2–3 | 64/126 (50.8%) |
| 4–6 | 29/126 (23.0%) |
| 7–9 | 4/126 (3.2%) |
| 10+ | 6/126 (4.8%) |

[a]"On-call duty" refers to moments when practitioners are not present at the clinic but can be called for a specific emergency and have to come in to assess patient or perform emergency surgery, whether during business hours or not.
[b]Any frequency between one week/one day out of three, two weeks/two days out of three, or 100% of the time.
[c]Any other frequency then those mentioned in the questionnaire.

offer pamphlet or other information explaining anaesthesia procedure and related risk than referral centres (69% (78/113) *vs*. 31% (4/13), *P* = 0.011). Potential risks during anaesthesia are explained by the receptionist, the animal health technician or the veterinarian in 23% (19/82), 67% (55/82) and 62% (51/82) of cases, respectively. An informed consent form is provided to and signed by the owner in 96% (120/125) of the practices.

**Pre-anaesthetic fasting.** Nearly all respondents (98%, 105/107) fast healthy patients for 6 to 12 hours prior to anaesthesia in small animals. Only 46% (50/108) of respondents give free access to water to healthy patients before anaesthesia.

Among respondents, 18% (19/106) do not fast paediatric patients, 29% (31/106) fast them for 4 hours or less, and 53% (56/106) for 6 to 12 hours before anaesthesia. Fifty-two percent (52%, 54/103) give free access to water to paediatric patients before anaesthesia.

**Table 2. Relations between risk factors of English-speaking veterinarians (n = 126) responding to a survey on management of anaesthesia in small animal practices in Canada.**

| Risk 1 | Risk 2 | *P*-value | Comments |
|---|---|---|---|
| Gender | Years since graduation | **0.01** | More men (69%) than women (45%) graduated more than 15 years ago |
| | Number of veterinarian(s) | 0.40 | |
| | On-call duty | 0.34 | |
| | Size of town | 0.76 | |
| | Type of practice | 0.24 | |
| | Number of animal(s) anaesthetised/day | 0.50 | |
| Years of practice since graduation | Number of veterinarian(s) | **0.006** | More respondents graduated less than 15 years ago work in large (5+ practitioners) rather than small team practices |
| | On-call duty | 0.14 | |
| | Size of town | 0.49 | |
| | Type of practice | 0.26 | |
| | Number of animal anaesthetised/day | 0.11 | |
| Number of animal(s) anaesthetised/day | Type of practice | **<0.001** | More animals are anaesthetised per day in referral centre than in general practice |
| | Number of veterinarian(s) | **<0.001** | More animals are anaesthetised in large (5+ veterinarians) rather than small team practices |

**Pre-anaesthetic evaluation.** Nearly all respondents (98%, 124/126) answered that a physical examination is performed in pre-anaesthetic evaluation for all patients, including paediatric (99%, 125/126), geriatric (99%, 125/126) and debilitated ones (99%, 125/126). The examination is performed in most cases within 24 hours before anaesthesia, both for routine surgeries (88%, 104/118) and for other surgeries (92%, 109/118). The parameters evaluated by respondents during physical examination are presented in Table 3.

Additional diagnostic tests are recommended by 69% (83/120) of the respondents for all patients, 71% (85/120) for paediatric patients, 90% (108/120) for geriatric patients and 95% (114/120) when they think it is necessary. Veterinarians graduated less than 15 years ago are more likely to recommend additional diagnostics for young patients than those graduated over 15 years ago (80% (44/55) *vs.* 63% (41/65), P = 0.047). The additional diagnostic tests recommended according to patient category are detailed in Table 4. Veterinarians graduated less than 15 years ago are more likely to recommend haematocrit and total protein measurement

**Table 3. Physical examination parameters evaluated by English-speaking veterinarian respondents (n = 120).**

| Physical examination parameter | Respondents performing it |
|---|---|
| Cardiac auscultation | 98% (117/120) |
| Thoracic auscultation | 95% (114/120) |
| Heart rate | 98% (117/120) |
| Respiratory rate | 90%, (108/120) |
| Temperature | 87% (104/120) |
| Abdominal palpation | 78% (93/120) |
| Lymph node palpation | 77% (92/120) |
| Peripheral pulse palpation concomitant to heart auscultation | 71% (85/120) |
| Patient history (including appetite, drinking, urination and defecation) | 93% (111/120) |
| All of the above | 60% (72/120) |

**Table 4. Additional diagnostic tests recommended by English-speaking veterinarians for each patient category.**

| Diagnostic test | Patient | Respondents recommending it |
|---|---|---|
| Haematocrit and total protein | Healthy | 55% (65/119) |
| | Paediatric | 55% (65/119) |
| | Geriatric | 49% (58/119) |
| | Believed at risk | 52% (62/119) |
| Haematology | Healthy | 69% (82/119) |
| | Paediatric | 63% (75/119) |
| | Geriatric | 92% (109/119) |
| | Believed at risk | 92% (109/119) |
| Hepatic enzymes | Healthy | 76% (91/119) |
| | Paediatric | 68% (81/119) |
| | Geriatric | 94% (112/119) |
| | Believed at risk | 94% (112/119) |
| Blood urea nitrogen and creatinine | Healthy | 78% (94/120) |
| | Paediatric | 73% (87/120) |
| | Geriatric | 93% (112/120) |
| | Believed at risk | 92% (110/120) |
| Glycaemia | Healthy | 62% (74/120) |
| | Paediatric | 67% (80/120) |
| | Geriatric | 78% (93/120) |
| | Believed at risk | 79% (95/120) |
| Urinalysis | Healthy | 13% (15/120) |
| | Paediatric | 10% (12/120) |
| | Geriatric | 54% (65/120) |
| | Believed at risk | 68% (81/120) |
| Electrocardiogram | Healthy | 3% (3/120) |
| | Paediatric | 2% (2/120) |
| | Geriatric | 18% (21/120) |
| | Believed at risk | 53% (64/120) |
| Radiography | Healthy | 3% (3/120) |
| | Paediatric | 1% (1/120) |
| | Geriatric | 17% (20/120) |
| | Believed at risk | 65% (78/120) |
| Electrolytes | Healthy | 33% (40/120) |
| | Paediatric | 35% (42/120) |
| | Geriatric | 67% (80/120) |
| | Believed at risk | 75% (90/120) |

Note: Grey-highlighted sections are indicated for their high occurrence rate.

For patients in good health, 28% (33/116) practitioners consider these procedures are accepted by at least 60% owners. For young patients, geriatric patients and patients believed to be at risk, 25% (29/115), 83% (97/117) and 81% (96/119) practitioners consider these procedures are accepted by at least 60% owners, respectively. Clients of GP respondents are less likely to accept recommended diagnostic tests for patients believed in good health compared to clients of respondents working in a referral centre (clients only having 0–20% chances saying yes to diagnostic tests were estimated at 29% (30/104) in first-line clinic *vs*. 0% (0/12) in referral centre, $P = 0.019$). American Society of Anesthesiologists (ASA) physical status classification is evaluated by 50% (57/115) of respondents for routine surgery, and by 51% (59/116) for non-elective surgeries.

for at-risk patients than those graduated over 15 years ago (63% (34/54) *vs*. 43% (28/65), *P* = 0.042). English-speaking GP veterinarians are more likely to recommend haematology (74% (78/106) *vs*. 31% (4/13), *P* = 0.003), hepatic enzyme (82% (87/106) *vs*. 31% (4/13), *P*<0.001) and blood urea nitrogen and creatinine evaluation (83% (89/107) *vs*. 38% (5/13), *P* = 0.001) for healthy patients than veterinarians working in a referral centre.

## Part III—Anaesthesia procedure

**Availability of emergency drugs.**   Overall, 28% (33/118) of respondents answer that they prepare emergency drugs before anaesthesia for all procedures, 43% (51/118) for procedures considered at-risk and 29% (34/118) never do. Ninety-three percent (93%, 110/118) of respondents have access to an emergency crash cart, with drugs and equipment for cardiopulmonary resuscitation. Among emergency drugs, 96% (111/116) of respondents use epinephrine, 94% (109/116) atropine, 73% (77/105) glycopyrrolate, 72% (76/106) doxapram, 39% (36/92) dopamine, 32% (30/94) dobutamine, 25% (22/89) ephedrine, 21% (19/89) vasopressin, and 10% (9/86) phenylephrine. Frequency of use for each drug is illustrated (see Fig 1), which shows that practices regularly use anticholinergic (atropine and glycopyrrolate) and catecholamine-like substance drugs. Practitioners working in referral centres are more likely to use phenylephrine (42% (5/12) *vs*. 5% (4/74), *P* = 0.002), ephedrine (67% (8/12) *vs*. 18% (14/77), *P* = 0.001), dobutamine (77% (10/13) *vs*. 25% (20/81), *P*<0.001), dopamine (75% (9/12) *vs*. 34% (27/80), *P* = 0.010), glycopyrrolate (100% (13/13) *vs*. 70% (64/92), *P* = 0.038) and vasopressin (75% (9/12) *vs*. 13% (10/77), *P*<0.001) than GPs.

Among respondents using drugs that could be antagonised, 93% (93/100) report to use naloxone, 79% (73/92) atipamezole, 40% (29/72) yohimbine, 16% (10/62) flumazenil and 11% (7/62) tolazoline).

**Premedication.**   Premedication is used by all respondents: 22% (24/111) use a premix (mix prepared ahead of time, same dosage for all patients), 7% (8/111) use the same protocol for all patients but mix drugs just before administration, and 71% (79/111) use individualised protocols, with different drugs and doses for each patient. The frequency of use of each drug for routine surgery is summarised in Fig 2. Briefly, non-steroidal anti-inflammatory drugs (NSAID), sedatives (dexmedetomidine and acepromazine), opioids (butorphanol and hydromorphone), and glycopyrrolate are commonly used for routine surgeries. Veterinarians graduated less than 15 years ago use hydromorphone in premedication of routine surgeries more often than older graduated respondents (81% (38/47) use it more than 20% of the time *vs*. 61% (30/49), *P* = 0.042). For premedication of routine surgeries, veterinarians working in GP are less likely to use midazolam (22% (16/72) use it more than 20% of the time *vs*. 67% (6/9), *P* = 0.023), but more likely to use glycopyrrolate (64% (52/81) use it more than 20% of the time *vs*. 11% (1/9), *P* = 0.036) than those working in referral centres.

The following drugs are used in premedication by respondents for non-routine surgeries: hydromorphone (93%, 85/91), butorphanol (91%, 84/92), glycopyrrolate/atropine (81%, 68/84), acepromazine (81%, 79/97), NSAID (72%, 56/78), diazepam (71%, 61/86), buprenorphine (71%, 57/80), fentanyl (71%, 51/72), dexmedetomidine (70%, 64/91), midazolam (64%, 47/74), morphine (48%, 32/67), medetomidine (29%, 18/63) and xylazine (18%, 11/62). For premedication of non-elective cases, morphine is more likely to be used in referral centres than in GPs (88% (7/8) *vs*. 42% (25/59), *P* = 0.023).

**Induction.**   The drugs used by respondents for induction for routine and non-routine surgeries are presented in Table 5. For induction of routine surgeries, veterinarians graduated less than 15 years ago are less likely to use alfaxalone (69% (22/32) use it in 0–20% cases *vs*. 55% (23/42), *P* = 0.047) and ketamine (94% (33/35) use it in 0–20% cases *vs*. 74% (31/42),

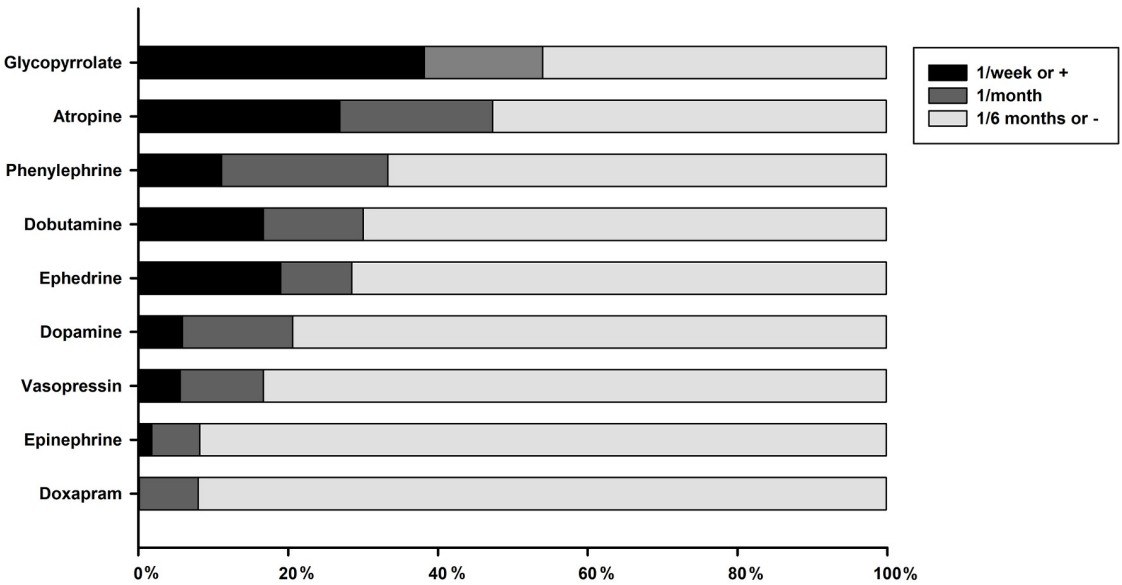

**Fig 1. Cumulative percentage of respondents reporting their frequency of use for each emergency drug in small animals anaesthesia.** Frequency of use is color-coded for at least 1/week, 1/month and 1/6 months or less.

$P$ = 0.025) than veterinarians graduated over 15 years ago. Veterinarians graduated over 15 years ago were more likely to use thiopental for induction of non-elective surgeries than younger veterinarians (41% (15/37) *vs*. 16% (5/32), $P$ = 0.03). Respondents working in referral centres use ketamine-medetomidine (75% (6/8) *vs*. 29% (16/56), $P$ = 0.016) or thiopental (67% (6/9) *vs*. 23% (14/60), $P$ = 0.014) more frequently for induction of non-elective surgeries than GPs.

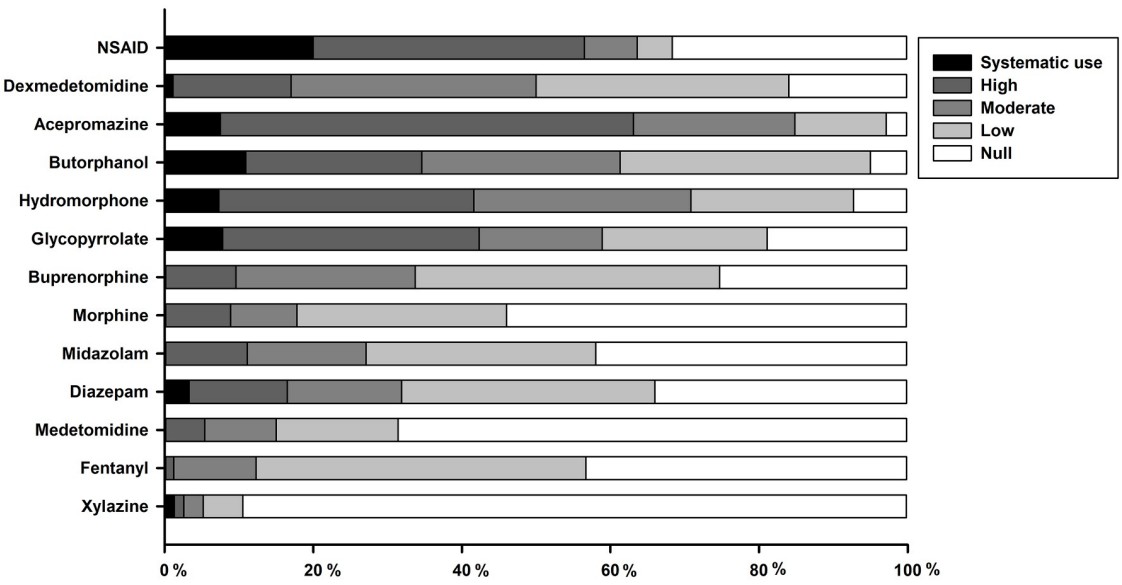

**Fig 2. Cumulative percentage of respondents reporting their frequency of use for each drug administered in small animals premedication for routine surgery.** Frequency of use is color-coded, as systematic (or 100%), high (61 to 99%), moderate (21 to 60%), low (1 to 20%) or null (or 0%).

Table 5. Drugs used by English-speaking veterinarian respondents for induction of routine and non-routine surgeries.

| Drug | Respondents using it for induction of routine surgery | Respondents using it for induction of non-routine surgery |
|---|---|---|
| Propofol | 92% (90/98) | 90% (84/93) |
| Ketamine combined with diazepam | 88% (84/95) | 85% (73/86) |
| Alfaxalone | 61% (45/74) | 59% (41/70) |
| Ketamine combined with (dex)medetomidine | 41% (28/69) | 34% (22/64) |
| Thiopental | 32% (22/69) | 29% (20/69) |
| Ketamine alone | 29% (22/77) | 25% (17/68) |

**Maintenance.** Anaesthesia with injectable agents alone is performed by 38% (42/112) of respondents. Veterinarians graduated less than 15 years ago are more likely to use injectable anaesthesia for maintenance than older veterinarians (49% (25/51) *vs.* 28% (17/61), $P = 0.031$). Respondents in referral centres are more likely to perform injectable anaesthesia than GPs (73% (8/11) *vs.* 34% (34/101), $P = 0.019$). Drugs used for maintenance include: propofol (74%, 29/39), ketamine (33%, 13/39), and alfaxalone (31%, 12/39). Anaesthesia with injectable agents alone is mostly used (96%, 50/52) for procedures considered rapid to perform and mildly painful by the respondents such as handling, castration of a male cat, skin biopsy, porcupine quills removal, or bronchoscopy.

When using inhalant anaesthesia, 100% (110/110) of respondents use isoflurane, 5% (5/110) use sevoflurane and 2% (2/110) use nitrous oxide.

**Anaesthesia machine.** Among respondents using inhalant anaesthesia, 97% (108/111) possess a Bain circuit (modified Mapleson D) and 95% (106/111) a rebreathing system. Therefore, 6% (5/111) possess only a Bain circuit and 3% (3/111) possess only a rebreathing system.

**Analgesia.** Regarding analgesia, 2% (3/125) of respondents consider that patients rarely need analgesia after surgery. Thirty-two percent (40/126) of respondents offer analgesia protocol as optional and the receptionist is the one discussing this option in 10% (4/40) cases, whereas the animal health technician or the veterinarian is discussing it in 55% (22/40) and 63% (25/40) cases, respectively.

All respondents use NSAID for routine surgeries: 54% (61/112) during recovery, 16% (18/112) at the same time as premedication, 11% (12/112) during surgery before the incision, 19% (21/112) during surgery but after the incision. After surgery, 76% (84/110) use NSAID for 3 to 4 days, 2% (2/110) for 7 days, and 22% (24/110) administer only a single NSAID injection peri-operatively. If NSAIDs are used, the respondents' preferred NSAIDs in dogs and cats for post-anaesthetic analgesia are reported in Table 6, with meloxicam being the most popular in both canine and feline patients.

Among respondents, 98% (107/109) use opioids after surgery: 18% (20/109) administer a single injection, 57% (62/109) only administer opioids as needed, 23% (25/109) administer systematically one dose after surgery and repeat as needed and 2% (2/109) never use opioid post-surgery. The respondents' preferred opioids in dogs and cats for post-anaesthetic analgesia are reported (Table 6), with hydromorphone and buprenorphine being the most popular in canine and feline patients, respectively. Amongst opioids used for post-operative analgesia in dogs, hydromorphone is the most commonly used, but veterinarians graduated less than 15 years ago use hydromorphone even more frequently over other opioids compared to veterinarians graduated over 15 years ago (86% (42/49) *vs.* 64% (32/50), $P = 0.02$). Opioids and NSAID are used together by 90% (99/110) of respondents.

Forty two percent (42%, 47/111) of respondents provide analgesia as an intravenous infusion during surgery. Respondent working in referral centres are more likely to use constant

**Table 6. English-speaking veterinarians' preferred NSAID and opioid in dogs and cats for post-surgery analgesia.**

| | Dog | Cat |
|---|---|---|
| **NSAIDs** | | |
| Meloxicam | **79% (85/109)** | **81% (88/109)** |
| Carprofen | **13% (14/108)** | 3% (3/109) |
| Tolfenamic acid | 2% (2/108) | **7% (8/109)** |
| Deracoxib | 3% (3/108) | 0% (0/109) |
| Firocoxib | 1% (1/108) | 0% (0/109) |
| Ketoprofen | 3% (3/108) | 6% (7/109) |
| Robenacoxib | 0% (0/108) | 3% (3/109) |
| **Opioids** | | |
| Hydromorphone | **75% (74/99)** | **26% (27/103)** |
| Buprenorphine | **11% (11/99)** | **64% (66/103)** |
| Butorphanol | 8% (8/99) | 9% (9/103) |
| Morphine | 6% (6/99) | 1% (1/103) |

Note: The two most frequently used drugs in each species are in bold.

rate infusion of analgesics (100% (11/11) *vs.* 36% (36/100), $P<0.001$) than respondents working in GP. The drugs most frequently used are ketamine (91%, 43/47), lidocaine (66%, 31/47), and fentanyl (49%, 23/47). Fentanyl is used in infusion more often in referral centres than in GPs (91% (10/11) *vs.* 36% (13/36), $P = 0.002$). Seventy-eight (78%, 86/110) of respondents use locoregional analgesic techniques. The techniques used most frequently are ring block for declawing (78%, 67/86), mandibular (73%, 63/86), maxillary (71%, 61/86), infra-orbital (58%, 50/86), and mental (49%, 42/86) nerve blocks. Thirty-three percent (34%, 29/86) of respondents answered performing other type of nerve blocks, among which infiltrative incisional line and intratesticular blocks are the most frequent. Respondents working in referral centres are more likely to use infra-orbital (90% (9/10) *vs.* 54% (41/76), $P = 0.04$) and mental nerve blocks (80% (8/10) *vs.* 45% (34/76), $P = 0.047$) than GPs.

## Part IV—Monitoring and safety

Technical procedures performed for anaesthesia are summarised for dogs (see Fig 3) and cats (see Fig 4). There are similarities in these anaesthetic acts both in dogs and cats, but endotracheal intubation and intravenous catheterisation are more frequent in the dog than in the cat. Systematic use of fluid therapy and preoxygenation is infrequent, in particular in cats. Respondents graduated less than 15 years ago are more likely to pre-oxygenate dogs than those graduated over 15 years ago (52% (26/50) do it in more than 20% cases *vs.* 31% (17/54), $P = 0.032$). Veterinarians working in referral centres are more likely to pre-oxygenate dogs than those working in GPs (73% (8/11) pre-oxygenate in more than 20% cases *vs.* 38% (35/93), $P = 0.028$).

When performing anaesthesia with injectable drugs only, respondents provide oxygen to the patient using a mask (22%, 22/100), using endotracheal intubation connected to an anaesthetic machine (43%, 43/100), by placing the oxygen supply in front of the patient nose (4%, 4/100) and 31% (31/100) do not provide oxygen to the patient. Respondents graduated over 15 years ago are more likely to provide oxygen *via* endotracheal intubation during injectable anaesthesia compared to those graduated more recently (57% (30/53) *vs.* 28% (13/47), $P = 0.011$). During injectable anaesthesia, all respondents working in referral centres (10/10) give oxygen supplementation whereas 66% (59/90) respondents working in GP do ($P = 0.012$).

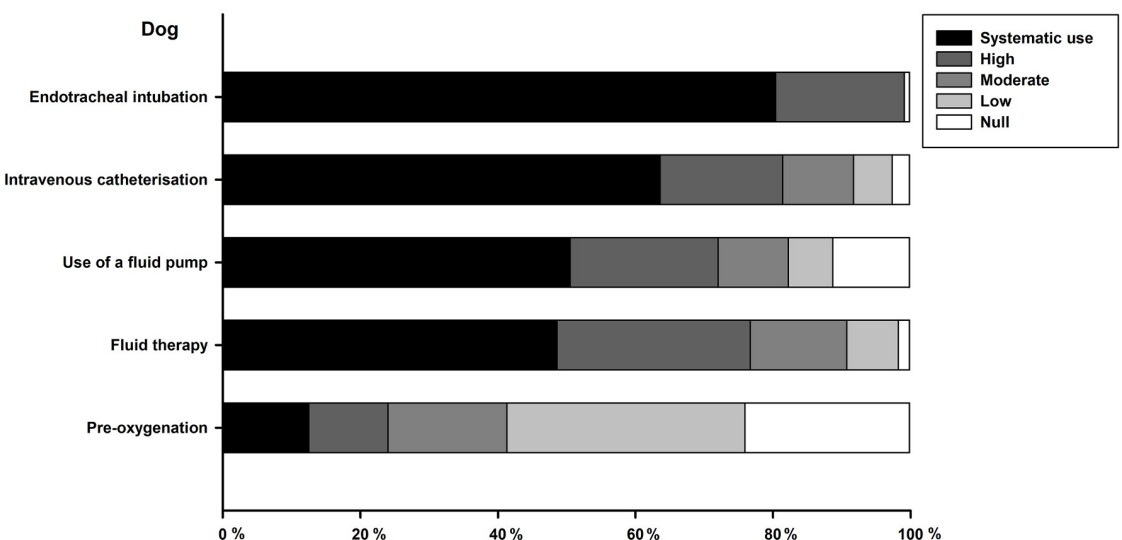

**Fig 3. Cumulative percentage of respondents reporting their frequency of use for each technical procedure performed for dog anaesthesia.** Frequency of use is color-coded, as systematic (or 100%), high (61 to 99%), moderate (21 to 60%), low (1 to 20%) or null (or 0%).

Parameters used to monitor cardiovascular, respiratory and neurological functions are presented in Table 7. Respondents graduated less than 15 years ago are more likely to use an electrocardiogram (ECG) than those graduated over 15 years ago (63% (32/51) *vs.* 41% (24/58), $P = 0.035$). All respondents working in referral centres monitor cardiovascular function with ECG, but not all respondents do in GPs (100% (11/11) *vs.* 46% (45/98), $P<0.001$). Sixty-four percent (64%, 70/109) of respondents use a device to monitor the respiratory rate. Significantly more respondents working in referral centres use capnography compared to those working in GPs (82% (9/11) *vs.* 33% (32/98), $P = 0.002$).

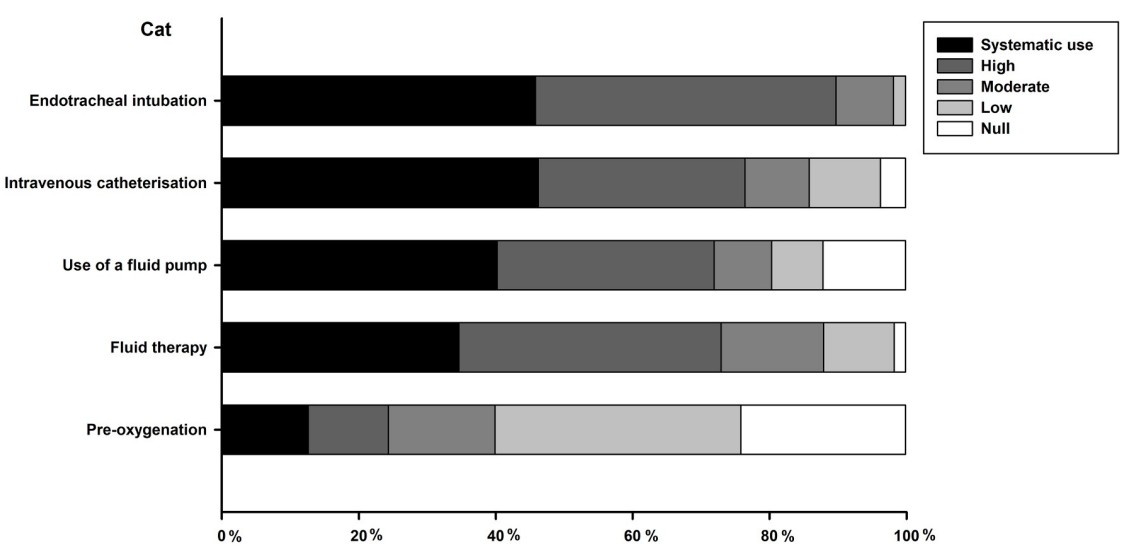

**Fig 4. Cumulative percentage of respondents reporting their frequency of use for each technical procedure performed for cat anaesthesia.** Frequency of use is color-coded, as systematic (or 100%), high (61 to 99%), moderate (21 to 60%), low (1 to 20%) or null (or 0%).

**Table 7. Parameters assessed to monitor cardiovascular, respiratory and neurological functions by English-speaking veterinarians responding to the survey.**

| Function | Parameter | Respondents assessing it |
|---|---|---|
| Cardiovascular | Heart rate | 97% (106/109) |
| | Mucous membrane colour and capillary refill time | 88% (96/109) |
| | Systemic arterial blood pressure | 82% (89/109) |
| | Cardiac auscultation | 68% (74/109) |
| | Peripheral pulse | 58% (63/109) |
| | Electrocardiogram | 51% (56/109) |
| Respiratory | Respiratory rate | 90% (98/109) |
| | Pulse oximetry | 84% (92/109) |
| | Lung auscultation | 55% (60/109) |
| | Capnography | 38% (41/109) |
| Neurological | Palpebral reflex | 97% (105/108) |
| | Jaw tone | 93% (100/108) |
| | Eye position | 90% (97/108) |
| | Pharyngeal reflex | 72% (78/108) |
| | Withdrawal reflex | 60% (65/108) |

Availability and use of monitoring devices by respondents are reported in Table 8. Apnea monitor is used more frequently by veterinarians graduated over 15 years ago in routine cases compared to those graduated less than 15 years ago (85% (17/20) *vs.* 50% (6/12), $P = 0.049$). ECG is used more often in referral centres for both routine and non-elective cases (90% (9/10) *vs.* 46% (33/72), $P = 0.015$; 100% (10/10) *vs.* 65% (47/72), $P = 0.028$, respectively).

When needed, complementary exams can be performed during the procedure by 95% (104/109) of the respondents. Respondents graduated less than 15 years ago are more likely to have access to in-house haematology (96% (47/49) *vs.* 67% (37/55), $P<0.001$), biochemistry (96% (47/49) *vs.* 80% (44/55), $P = 0.017$) and electrolytes (88% (43/49) *vs.* 60% (33/55), $P = 0.002$) than those graduated over 15 years ago. Referral centres are much more likely to have all mentioned additional diagnostics and laboratory exams readily accessible compared to GPs (91% (10/11) *vs.* 3% (3/93), $P<0.001$). This includes blood gas analysis (91% (10/11) *vs.* 10% (9/93), $P<0.001$), blood typing (91% (10/11) *vs.* 8% (7/93), $P<0.001$) and crossmatching (91% (10/11) *vs.* 16% (15/93), $P<0.001$), individually as well.

**Table 8. Use of monitoring devices by English-speaking Canadian veterinarians having access to mentioned monitoring device.**

| Monitoring device | Used in routine cases | Used in non-routine cases | Available in the clinic, but not used |
|---|---|---|---|
| Pulse oximeter | 89% (83/93) | 78% (73/93) | 9% (8/93) |
| Doppler blood pressure | 72% (46/64) | 77% (49/64) | 16% (10/64) |
| Electrocardiogram | 51% (42/82) | 70% (57/82) | 27% (22/82) |
| Oscillometric blood pressure | 80% (49/61) | 77% (47/61) | 11% (7/61) |
| Multi-parametric monitor | 84% (47/56) | 82% (46/56) | 11% (6/56) |
| Capnograph/Capnometer | 79% (38/48) | 71% (34/48) | 15% (7/48) |
| Apnea monitor | 72% (23/32) | 63% (20/32) | 25% (8/32) |
| Oesophageal stethoscope | 29% (20/68) | 40% (27/68) | 56% (38/68) |
| Blood gases analyser | 18% (3/17) | 82% (14/17) | 18% (3/17) |
| Invasive blood pressure | 9% (1/11) | 73% (8/11) | 27% (3/11) |

Transfusion is not an option for 61% (66/109) of the respondents. Respondents working in referral centres are more likely to be able to perform blood transfusion compared to GPs (91% (10/11) *vs.* 34% (33/98), *P*<0.001).

A ventilator is available for 19% (21/109) of respondents. Respondents working in referral centres are more likely to have a mechanical ventilator compared to GPs (91% (10/11) *vs.* 11% (11/98), *P*<0.001). In GPs, mechanical ventilation is never used in 64% (7/11) respondents.

During routine surgeries, monitoring is performed by someone dedicated to this task (69%, 74/108), someone helping with the surgery (29%, 31/108) or the person doing the surgery (3%, 3/108). During non-routine surgeries, monitoring is performed by someone dedicated to this task (77%, 84/109), someone helping with the surgery (20%, 22/109) or the person doing the surgery (3%, 3/109). During non-elective surgery anaesthesia, monitoring is performed by a dedicated staff member more often with respondents graduated over 15 years ago than those graduated more recently (83% (48/58) *vs.* 71% (36/51), *P* = 0.020).

Monitoring data are systematically recorded on an anaesthesia record by 77% (84/109) of the respondents, and never by 23% (25/109). Recording is reported to be performed every 5 min (71%, 77/109), every 10 minutes (8%, 9/109) or at no specific interval (21%, 23/109). All respondents who answered that anaesthetic monitoring is performed at no fixed frequency are working in GPs and all respondents working in referral centres perform monitoring at a specific frequency (23% (23/98) *vs.* 0% (0/11), *P* = 0.03).

During recovery, monitoring of the patient include visual monitoring (eye position, mucous membrane colour, thoracic movements– 98%, 107/109), temperature (79%, 86/109), tactile monitoring (pulse quality, jaw tone, palpebral reflex– 87%, 95/109), auscultation (78%, 85/109) and the same monitoring as during anaesthesia (10%, 11/109).

Monitoring during recovery is continued until the patient is able to remain in sternal recumbency (70%, 76/109), the patient temperature is considered normal (31%, 34/109) and/ or the patient is extubated (43%, 47/109). Respondents graduated less than 15 years ago monitor the animals during recovery until they have reached normal body temperature more often than those graduated over 15 years ago (41% (21/51) *vs.* 22% (13/58), *P* = 0.04). For routine surgery, respondents stop rewarming the patient when its rectal temperature reaches 36˚C (2%, 2/108), 37˚C (32%, 35/108) or 38˚C (44%, 48/108). Twenty one percent (21%, 23/108) of the respondents do not always measure temperature during recovery. Means of warming up patients include hot water heating mats (60%, 67/111), forced air warmer (38%, 42/111), electric plates/mats (35%, 39/111), and fluid heater (25%, 28/111). Hot therapeutic oat bags (17%, 19/111) and heating lamps (14%, 15/111) are used less often.

After routine surgery, 28% (30/109) respondents keep the patient hospitalised between 12 to 24 hours after surgery, 33% (36/109) for 6 to 12 hours after surgery, 1% (1/109) more than 24 hours after surgery and 39% (42/109) less than 6 hours after surgery.

## Discussion

This study describes current standards of small animal anaesthesia by English-speaking Canadian veterinarians and assesses how demographic factors (type of veterinary practice, number of animals anaesthetised per day, as well as the veterinarian's gender and experience) influenced the way anaesthesia is performed.

As was observed for French-speaking practitioners in a previous study [6], surveyed English-speaking practices of Canada do not generally follow the guidelines published, notably by the American Animal Hospital Association (AAHA) and the American College of Veterinary Anesthesia and Analgesia (ACVAA) [1–5, 9].

Evaluation and preparation of the patient appear to be sub-optimal. Current recommendation regarding fasting is to withhold food for 4 to 6 hours prior to anaesthesia for healthy adult patients or for 3 to 4 hours in cats [5, 10]. Dogs and cats less than 8 weeks old should not be fasted for more than 1 to 2 hours [1, 5]. Nearly all surveyed practitioners fast healthy adult animals 6 to 12 hours prior to anaesthesia, which may be longer than needed in several cases. It is worth noting that a 2015 reference recommends fasting healthy animals at least 6 hours prior to anaesthesia [11]. The recommended fast duration has decreased in recent years based on clinical experience and experimental evidence showing a lower incidence of gastroesophageal reflux [5]. Practitioners might use books already available at their clinic and may not be aware of the free access to regularly updated online guidelines, for example AAHA's [5]. Such outdated practice suggests that veterinary practitioners might find it difficult to keep abreast of the latest developments.

Water can be allowed until just prior to anaesthesia, unless the patient is at risk for regurgitation [1, 5, 11]. Based on these recommendations, 54% of the respondents do not meet the criteria for withholding water in healthy adult patients, and a similar percentage of respondents indicate no updated practice for pre-anaesthesia fasting of paediatric patients [5]. In the early 2000s, it was recommended to allow free access to water until up to 2 hours [12], 2–4 hours [13], or at least 2 hours before anaesthesia [14]. AAHA has recommended to give free access to water up to the time of premedication at least since 2011 [2]. Even though the guidelines have changed over the past decades, a significant proportion of practitioners withdraw water in all patients 6 to 12 hours prior to anaesthesia, which is excessive, even compared to older recommendations. Withdrawing water several hours before anaesthesia might cause dehydration and hypovolemia and puts patients at risk for hypotension. Again, this highlights that some practices failed to update their standards and are several years behind regarding pre-anaesthesia fasting recommendations.

Among respondents, almost all answered they performed physical examination prior to anaesthesia, but only 60% of respondents evaluate all physical parameters and obtain a history. This remains a worrying trend as reported in a recent survey [6]. Indeed, gathering a complete physical exam and history is recommended to orient additional diagnostic test requirements and avoid adverse drug interaction if the patient is already taking medication [1, 5, 10]. Furthermore, it has been reported that failure to record a physical exam increases the odds for death in dogs [15].

Only half the respondents recommend performing haematocrit and total protein for healthy, paediatric, geriatric and patients believed at risk. For patients in good health, very few practitioners consider these procedures are accepted most owners. The reluctance of owners to accept additional diagnostics may discourage practitioners to recommend them at all, especially if they are seemingly healthy. Whereas it has been reported that diagnostic tests can detect significant changes unsuspected based on physical exam and history in 6.2% of dogs and 19.2% of cats, some studies determined that if history and clinical examination did not report potential issues, pre-anaesthetic blood screening does not bring additional important information and does not change anaesthetic management [1, 16]. Indeed, over the years, there has been controversy on the matter and the need for pre-anaesthetic bloodwork in healthy patients has been questioned [16]. In human anaesthesia, consensus is that healthy patients undergoing elective procedures do not benefit from pre-anaesthetic bloodwork, but there is not yet agreement in veterinary medicine [16]. In the current study, more younger graduates recommend bloodwork for at-risk patients compared to older graduates, which seems justified, but they also are more likely to recommend tests for young patients. Veterinarians working in GPs recommend more blood tests in healthy patients than those working in

referral centres. Potential reasons could be to reassure oneself or objectively document the patient's health status prior to an intervention in the advent of a complication or lawsuit.

Clients of respondents working in GP are less likely to accept recommended diagnostics for all patients compared to clients of respondents working in a referral centre, suggesting a more motivated clientele in the latter. Furthermore, clients consulting in referral centre might be more likely to have the financial means to afford these tests. Another hypothesis is that if veterinarians working in referral centre recommend useful tests specific to the patients' condition and properly justify their usefulness, their clients are more likely to have them performed.

About half of the respondents evaluate ASA physical status grade for elective and non-elective surgeries. ASA is a prognostic tool that helps determine the need for stabilisation and predict the relative risk for mortality under anaesthesia [5, 17]. One feline study determined ASA physical status was a better predictor of perianaesthetic complications than age [10]. Veterinarians should take the time to properly assess anaesthetic risk for each patient, allowing them to address certain conditions preanaesthetically, to be prepared for potentially expected complications and treat them accordingly, thereby improving anaesthesia safety and patient outcome [16].

Most respondents have access to an emergency crash cart, but almost a third never prepare emergency drugs. It has been shown that the availability of emergency carts and drugs affects the outcome of cardiopulmonary resuscitation [5]. Among cardiopulmonary resuscitation complications, incorrect emergency drug dosages are frequently reported [18]. Therefore, one should have emergency equipment and drugs readily available and doses calculated [1].

Premedication is used by all respondents, but almost a third do not use individualised protocols. The goal of premedication is to reduce patient's anxiety, decrease doses of other induction and maintenance drugs and provide analgesia. Therefore, it should be tailored to each patient and procedure [1, 5, 9, 10]. With only 71% of respondents using individualised anaesthesia / analgesia protocol, the risk of an inadequate analgesic plane is high with premixes. Xylazine has been associated with increased mortality in dogs and cats [3, 19, 20]. There are still 18% respondents that use it for premedication of non-routine surgeries.

A few respondents possess only a Bain circuit or only a rebreathing system. These respondents may not be able to anaesthetise all sizes of patients properly. Indeed, nonrebreathing circuits such as Bain circuit are often recommended for small patients (<3–5 kg) as they may decrease resistance to breathing and dead space, lowering the risk of $CO_2$ rebreathing [5]. Some rebreathing circuits can be used in these small patients only if paediatric rebreathing circuit is available [5]. It is also suboptimal to use Bain circuit with large patients which will consume high amounts of oxygen and anaesthetic gas and be at risk of re-inspiration.

Despite guidelines [3–5], about a third of respondents still present analgesia as an option for owners of patients undergoing routine surgery. Pain management is vital for all patients undergoing surgery. Indeed, unrelieved pain can have deleterious long-term consequences on the patient such as maladaptive physiological responses and behaviours and may lead to pathological pain [4, 9, 21]. Veterinarians have a professional obligation of ensuring animals' welfare and no procedure should be performed without adequate pain management [4].

All respondents use NSAIDs for routine surgery and half administer them during recovery. In Canada, Metacam®, Onsior™ and Rimadyl® amongst others are homologated for perioperative pain management with the first injection given before the surgery [22–24]. Additionally, NSAIDs might be more efficient when given prior to a painful procedure as preemptive analgesia [3, 25, 26]. Fear of potential nephrotoxicity if hypotension occurs during the anaesthetic episode might explain why veterinarians tend to administer them at the end of anaesthetic episode [3, 4]. Indeed, if normotension cannot be ensured, it was recommended by AAHA and American Association of Feline Practitioners (AAFP) Task Force to perform NSAID

administration after the surgery [4]. This seems to imply that some veterinarians are not confident that adequate blood pressure monitoring, and maintenance will be achieved during routine surgery.

Constant rate infusion of analgesic agents can provide multimodal analgesia and anaesthesia during induction, maintenance and recovery period and allows a decrease in inhalant anaesthetic concentration needed [5, 10]. The goal of multimodal analgesia is also effective pain management by targeting several sites in pain pathway and decreasing the risk of side effects by lowering doses of each drug [4]. Analgesia provided as a constant rate infusion is used significantly more frequently in referral centres compared to GPs. Perhaps procedures done in GP setting are considered too short to be worth preparing a constant rate infusion. Drug dilution and infusion rate calculation might be a challenge for some, discouraging its use. Continuing education might help veterinarians working in GPs to learn about this modality.

Another way to provide multimodal analgesia is with locoregional analgesic techniques, which are used by most respondents, as encouraged for all surgeries by current guidelines for their safety and significant benefits [4, 5]. Several local blocks (for example infiltration blocks or splash blocks) are easy to perform, efficient and inexpensive, therefore there is no reason why a veterinarian should not use them, except lack of proper training.

There are several worrying results regarding patient monitoring and safety. Only 64% and 46% of English-speaking practitioners always place intravenous catheterisation for general anaesthesia of dogs and cats, respectively. Current guidelines state intravenous catheter placement is mandatory in almost all situations including very short procedures to benefit from ease to administer additional anaesthetic, analgesic or emergency drug and fluid therapy [1, 5, 10]. Endotracheal intubation is more frequent in the dog than in the cat with less than half respondents that always intubate cats. Perhaps it is because cats may be more difficult to intubate and are often anaesthetised with injectable agents only, namely for castration [5]. Complications related to endotracheal intubation were associated with anaesthetic-related deaths in cats as well [19, 27]. Despite this, endotracheal intubation is essential to maintain airways open and protected from aspiration, and allows mechanical ventilation [5, 10]. It has been stated that the delivery of oxygen without an endotracheal tube may be preferable for short, minor procedures in cats, but significant advantages of intubation cannot be neglected and overcome the risks when performed properly otherwise [10, 27]. One should refer to AAFP Anesthesia Guidelines for atraumatic intubation tips in cats [10]. Other possible explanations for infrequent endotracheal intubation in feline patients such as technical or time limitations in high-volume practices should be investigated.

Systematic use of fluid therapy and preoxygenation is infrequent, particularly in cats. Preoxygenation is an integral part of pre-anaesthetic / induction sequence and should be done in most cases [1, 5, 10]. Balanced crystalloid fluids are beneficial for most patients undergoing anaesthesia except for very short procedures [5, 10]. Intravenous fluid administration in cats has sometimes been associated with increased odds of anaesthesia-related death, but there were potential confounding risk factors [27]. In addition, guidelines have changed over the years, with more conservative fluid rates recommended now [5, 28]. Indeed, recommended basal fluid rate changed from 10 mL/kg/h to 5 mL/kg/h for dogs and 3 mL/kg/h for cats in 2013 [28]. Procedures done in cats might be considered too short to deserve fluid support or practitioners might fear fluid overload or occult cardiac disease in these small patients [10].

For anaesthesia with injectable drugs only, about a third of survey respondents do not provide oxygen to patients, which goes against AAHA recommendations [5]. Conversely, this is thoroughly applied in referral centres in which oxygen is always supplemented during injectable anaesthesia.

In this study, only 38% use capnography to monitor respiratory function and half of respondents use ECG on routine cases with a significant proportion of respondent having monitoring equipment available but not using it. A difference is again seen in referral centres where both capnography and ECG are used more often than in GPs. Adequate monitoring allows early detection of complications and is a way to mitigate risk of anaesthesia and decreases the odds of anaesthetic death [5, 10].

Anaesthetic record is not always used systematically and many respondents stop anaesthetic monitoring when the animal is extubated despite recommendations to document patient parameters during anaesthesia and recovery by AAHA and ACVAA and by several provincial governing bodies [5, 9, 29–31]. Roughly half of anaesthesia-related deaths occur during the recovery period, most frequently during the first 3 hours, therefore one should not underestimate the value of continuous monitoring even after extubation [32]. When performed during anaesthesia, most respondents record parameters at 5–10 min intervals as recommended by AAHA and ACVAA, but this recommendation was not reiterated in AAHA's most recent guidelines [2, 5, 9]. AAFP recommended to record parameters at least every 15 minutes in cats, although greater frequency allows better assessment of changes [10]. Twenty one percent (21%) of the respondents do not always measure temperature during recovery despite ACVAA and AAFP recommendations [9, 10]. Most common equipment used by respondents to warm patients are hot water heating mats and forced air warmer, which are the most effective to do so [5, 10].

Several other factors might explain the difference between published guidelines and the actual way anaesthesia is performed by English-speaking Canadian practitioners. In order to be competitive, some veterinarians may offer several "optional features" to clients, including post-op analgesia and monitoring. Even though it can be tempting to leave some decisions up to the client in order to make services affordable, analgesia quality should never be optional. Despite colleges of veterinary medicine attempting to provide optimal education considering recent guidelines, a sad truth is that recent veterinarian graduates receive clinical formation in their workplace and adhere to protocols already used by the veterinarians working there [33]. The latter might not be up to date in their formation and changing already well-in-place protocols can be challenging.

In addition to the previously mentioned influences of type of practice and years of experience on how anaesthesia is performed compared to guidelines, others were noted, namely drug and equipment availability and use. Referral centres are more likely to use emergency / vasopressor drugs regularly compared to GPs. This may be due to a better accessibility and continuing education given at the clinic or a greater exposition to cases requiring critical care. Injectable agents for maintenance of anaesthesia is used more often by veterinarians graduated less than 15 years ago and working in referral centres, mostly for short, mildly painful procedures. Referral centres are much more likely to have access to additional diagnostics, laboratory exams, mechanical ventilation and be able to perform blood transfusions compared to GPs. This again may illustrate more financial resources and higher caseload needing critical care, justifying the investment in such equipment. Respondents graduated less than 15 years ago are better equipped with several in-house blood tests. Overall, there seems to be an improvement in some anaesthetic practices in more recent veterinarian graduates compared to older respondents, including pre-anaesthetic evaluation, pre-oxygenation, ECG use, and access to additional diagnostics.

There are several limitations to consider in this survey. Selection bias is possible, and our sample might not be fully representative of the studied population, even if the demographic characteristics (see Table 1) of our sample look close to those of the general population [34] suggesting a good representativeness of the sample. Veterinarians answering the survey might

have a specific interest in anaesthesia, which can affect the results. The response rate was variable along the survey with more complex and later questions having fewer answers. Considering the potential number of responses that could be collected, the response rate to the questionnaire used to collect the data was 12.4% (126 respondents out of 1 016 sent invitations to small animal practitioners). However, a total of 189 veterinarians visited the questionnaire webpage and 126 of them provided a comprehensive set of responses for analysis, representing a response rate of 67% among those who showed interest. Although the first rate may appear to be low at first glance, it is well recognised that, on average, a rate of 10–15% is usually obtained in external surveys [35]. Response rates have historically been the method of choice for documenting survey quality and many journals require authors to report the response rates associated with their surveys. There has been a general lack of consensus regarding best practices for defining and calculating response rates, and there is no scientifically proven minimally acceptable response rate [36, 37]. The representativeness of the sample is much more important than the response rate [38]. The potential bias caused by the non-response rate cannot be ignored, but it does not make it possible to judge further the quality of the representativeness of the data collected [36]. Some results should be interpreted cautiously because of our limited power of analysis, and the difference in respondents sample size in each group (for example, the lower number of respondents working in referral centers compared to the higher number of respondents working as GPs). Further studies should be done with more respondents to confirm some findings. Finally, prospective studies on anaesthesia complication rates and outcome are needed to determine if diverging from guidelines impacts significantly the quality of animal care.

In conclusion, a proportion of surveyed Canadian English-speaking veterinarians do not follow several current small animal anaesthesia / analgesia guidelines. Veterinarian's experience and type of practice influenced anaesthesia management with practitioners working in referral centres closer to meet recommendations in general. Guidelines should be easily accessible in all veterinary practices and continuous education encouraged to better respond to these standards of care.

## Supporting information

**S1 Appendix. Questionnaire.** Presentation of the questionnaire used for the electronic survey, with the different sections, and all questions.
(DOCX)

**S1 File. Data responses to questions 8 to 27 of the survey.**
(XLSX)

**S2 File. Data responses to questions 28 to 45 of the survey.**
(XLSX)

**S3 File. Data responses to questions 46 to 66 of the survey.**
(XLSX)

## Acknowledgments

The authors wish to thank Dispomed Inc. (www.dispomed.com), and in particular Mrs Mélissa Lachapelle, for their technical assistance and active contribution to the success of the survey.

## Author Contributions

**Conceptualization:** Sophie Lalonde, Geoffrey Truchetti, Colombe Otis, Guy Beauchamp, Eric Troncy.

**Data curation:** Guy Beauchamp.

**Formal analysis:** Geoffrey Truchetti, Colombe Otis, Guy Beauchamp, Eric Troncy.

**Funding acquisition:** Eric Troncy.

**Investigation:** Sophie Lalonde, Geoffrey Truchetti, Colombe Otis.

**Methodology:** Colombe Otis, Guy Beauchamp, Eric Troncy.

**Project administration:** Colombe Otis, Eric Troncy.

**Resources:** Geoffrey Truchetti, Colombe Otis, Eric Troncy.

**Supervision:** Geoffrey Truchetti, Eric Troncy.

**Validation:** Geoffrey Truchetti, Colombe Otis, Eric Troncy.

**Writing – original draft:** Sophie Lalonde, Geoffrey Truchetti, Colombe Otis, Guy Beauchamp, Eric Troncy.

**Writing – review & editing:** Sophie Lalonde, Geoffrey Truchetti, Colombe Otis, Guy Beauchamp, Eric Troncy.

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
