## [Decision Letter · Decision Letter 0]

14 May 2021

PONE-D-21-05375

Management of veterinary anaesthesia and analgesia in small animals: A survey of English-speaking practitioners in Canada

PLOS ONE

Dear Dr. Troncy,

Thank you for submitting your manuscript to PLOS ONE. After careful consideration, we feel that it has merit but does not fully meet PLOS ONE’s publication criteria as it currently stands. Therefore, we invite you to submit a revised version of the manuscript that addresses the points raised during the review process.

We look forward to receiving your revised manuscript.

Kind regards,

Jan S Suchodolski, DVM, PhD

Academic Editor

PLOS ONE

Journal Requirements:

"There was not proprietary interest or funding directly provided for this project. This work was indirectly supported (ETR) by a Discovery grant (#441651–2013, supporting salaries) and a Collaborative Research and Development grant (#RDCPJ 491953–2016 supporting operations and salaries in partnership with ArthroLab Inc.) from the Natural Sciences and Engineering Research Council of Canada. COT is a recipient of a MITACS Canada Elevation postdoctoral scholarship (#IT11643). The authors got support from the company Dispomed Inc., i.e. to deliver the electronic survey to their clients. The specific roles of all authors are articulated in the ‘author contributions’ section. The funders had no role in study design, data collection and analysis, decision to publish, or preparation of the manuscript."

We note that you received funding from a commercial source: ArthroLab Inc.

4. Please include captions for *all* your Supporting Information files at the end of your manuscript, and update any in-text citations to match accordingly. Please see our Supporting Information guidelines for more information: http://journals.plos.org/plosone/s/supporting-information.

Reviewers' comments:

Reviewer's Responses to Questions

**Comments to the Author**

1. Is the manuscript technically sound, and do the data support the conclusions?

Reviewer #1: Partly

Reviewer #2: Yes

2. Has the statistical analysis been performed appropriately and rigorously? 

Reviewer #1: Yes

Reviewer #2: Yes

3. Have the authors made all data underlying the findings in their manuscript fully available?

Reviewer #1: Yes

Reviewer #2: Yes

4. Is the manuscript presented in an intelligible fashion and written in standard English?

Reviewer #1: Yes

Reviewer #2: Yes

5. Review Comments to the Author

Reviewer #1: Overall, I think this information is extremely useful for anesthesiologists to help guide our teaching practices and where to emphasize specific aspects of anesthesia within the curriculum. It is very eye-opening and important to our current knowledge base about veterinary anesthesia. Aside from the more specific comments below, I have a couple general comments to consider. One is I think I would consider altering the conclusion a bit as I don’t think your numbers truly represent the population of veterinarians. Perhaps consider stating a sig. proportion of those sampled, I think that would be more appropriate. In addition, the manuscript seems very lengthy. Perhaps consider using your tables more effectively to reduce the lengthy results section (e.g. state significance and p values in tables rather than text). I also think there are several areas in the discussion that would be worth condensing in larger categories rather than individually discussing each aspect of the results. Also, there are several results restated in the discussion that could probably be removed.

Abstract

Line 32: Due to the low response rate I am not sure I would use the term significant. Perhaps just state a proportion of veterinarians surveyed. Also, I think need you need to specify what the other “class” of veterinarians you are referring to. For example, perhaps specify general practitioners or primary care veterinarians are the ones that do not follow current guidelines.

Introduction

Line 49 – Please specify “Canada” rather than this country.

Materials and methods

Line 76: Please describe more as to what dispomed did as a contributor for survey distribution. It is sort of ambiguous7 at this point.

Results

Line 102: I think it is important to consider that your demographic data may be skewed because there is no information describing the types of practice that the survey was distributed too via dispomed. Therefore, I don’t believe it is “interesting” that majority of respondents were GPs in large or small towns because that may have been the predominant facilities in which the survey was delivered too. Perhaps just state the findings of the demographics without drawing too much conclusions from it.

Table 1

Can you better define the term on-call hours? I think you mean 5pm-8a/weekend? Also, what would you define as episodic? Lastly, can you specify what you mean by “years of graduation”. I think you mean years since veterinary school graduation. Years of graduation doesn’t really define anything.

Line 135: can you define what was considered pediatric? Since geriatric is used often we should define this.

Line 158: Hindsight... would have been interesting to identify the schools of graduation.

Line 161 – “blood” urea “nitrogen” evaluation

Table 3

What was classified or how was it determined for patients to be “believed at risk”?

“blood” urea “nitrogen and creatinine

Define the abbreviation ECG in the table

Line 238 – this goes for all areas where only percentages are shown for comparison. Since the number of respondents that answered that specific question differ, do you think it would be of interest to your readers that you provide them with the exact count for comparison rather than or in addition to percent? I understand it gets to be a bit cumbersome, however im not sure in this situation percentage really tells us an entire lot. Just for example purposes, I think 3/10 is way different than 30/100 despite both being 30%.

Table 4- please provide number of respondents for each item.

Line 273: what was the most common opioid for those graduated >15 years ago? would be interesting to report.

Line 279 – infusion not perfusion

Line 282 “nerve” blocks

Line 283 “nerve” blocks

Line 284 – “infiltrative incisional line” and “ intratesticular””

Line 285 – please make sure we use appropriate terminology “nerve block”

Table 5 – what do you mean present in the clinic? Is that number of respondents? Please provide the numbers in all percentages. Again because it depends on the number of respondents I think it would be more useful. Also for the tables I think the title should be more descriptive. It really doesn’t say much about the study. In my opinion it should be stand alone so the reader can view the table and understand the basics of the study.

Results in general – I appreciate the work you put into providing the vast amount of detail in the results. It is fairly lengthy and I am afraid you will begin to lose some of the readers because of its length (maybe I am wrong). Perhaps with better descriptors in your tables you can shift some of this information including p values and statistical significance for comparison in tables and reduce the text length.

Discussion

Line 433 can we separate this sentence up into 2? Also specify younger graduates are more likely when compared to who? We assume older graduates but please specify.

Line 439 – I get what you are saying but I do not think I would write that in an open access journal based on it being a big speculation.

Line467 – I don’t agree with this statement. Yes, it does cause increased resistance but not if positive pressure ventilation is provided (by hand or machine). In addition the studies used to assess resistance are fairly old and used metal valves which were heavy. Our university does not use NRB circuits just because of the concern a student will accidently hit the quick flush.

Line 475 – I think simon et al 2017 on oligoanalgesia would be a good reference for this section.

Line 480-481 – I think you need to support this with a reference. Many areas routinely give them postoperatively so perhaps a brief explanation as to why this is inappropriate in Canada.’

Line 490 – lets avoid restating the results in the discussion please.

Line 502 – inexpensive

line 508 – to administer “fluid”? be more specific

line 516- thanks for pointing this out

Line 521 – again too much results in the discussion. I understand there has to be some reference but in an already long manuscript I would try to reduce this when possible.

Line 535 – who’s recommendations?

line 602- please state a sig proportion of those surveyed… again I don’t think just over 100 vets is a sig portion of the population in Canada.

Reviewer #2: line 96: I am not sure weather " showed " is correct and it should be "shown" but of course English is not my mother tongue

I do not clearly understand 102-103

160-162 - quite surprising

173- a small tabel of ASA physical status classifications would be nice

476-477 cannot agree more!

524- could you discus in short the time it takes to preoxigenate animals, and that also might be a hustle in highly frequented clinics, but which benefits it gives.

562- absolutely

Congratulations on that very well written artikel. It is clear and comparable to the survey of french-speaking practitioners but more detailed. It points out that there is still much to be done in terms of providing safe anesthesia and meeting the current recommendations...

6. PLOS authors have the option to publish the peer review history of their article (what does this mean?). If published, this will include your full peer review and any attached files.

Reviewer #1: No

Reviewer #2: No

---

## [Author Response · Author response to Decision Letter 0]

7 Jul 2021

Response to the Reviewers’ comments on the manuscript

Title: Management of veterinary anaesthesia and analgesia in small animals: A survey of English-speaking practitioners in Canada

Ref: PONE-D-21-05375

Journal: PLOS ONE

Date: 2021-05-17

The authors would like to thank the Editorial Board and Reviewers for careful review of our manuscript and providing us with the comments to improve its quality. We have judiciously taken them into consideration in preparing our revision. The following detailed responses have been prepared to address the reviewer’s comments.

Reviewer #1

General comment:

Overall, I think this information is extremely useful for anesthesiologists to help guide our teaching practices and where to emphasize specific aspects of anesthesia within the curriculum. It is very eye-opening and important to our current knowledge base about veterinary anesthesia. Aside from the more specific comments below, I have a couple general comments to consider. One is I think I would consider altering the conclusion a bit as I don’t think your numbers truly represent the population of veterinarians. Perhaps consider stating a sig. proportion of those sampled, I think that would be more appropriate. In addition, the manuscript seems very lengthy. Perhaps consider using your tables more effectively to reduce the lengthy results section (e.g. state significance and p values in tables rather than text). I also think there are several areas in the discussion that would be worth condensing in larger categories rather than individually discussing each aspect of the results. Also, there are several results restated in the discussion that could probably be removed.

Response:

The authors thank the Reviewer for his/her general comment and rigorous revision about the present manuscript and the recognition that the main subject is of interest. Regarding the conclusion, we appreciate Reviewer #1’s comment and agree that we should not automatically extrapolate our results to the entire English-speaking Canadian population. Therefore, we modified the conclusion accordingly, so it involves specifically the respondents and not the entire population.

We also made an effort to condense results section by avoiding repeating P-values and significance in both the tables and text. Several tables were added to present results as well. The discussion was revised and modified to decrease repetition of results.

Finally, we made many other changes following your comments for some areas of the manuscript, leading to a more concise and complete revised manuscript. 

Minor point #1:

Line 32: Due to the low response rate I am not sure I would use the term significant. Perhaps just state a proportion of veterinarians surveyed. Also, I think need you need to specify what the other “class” of veterinarians you are referring to. For example, perhaps specify general practitioners or primary care veterinarians are the ones that do not follow current guidelines. 

Response:

We thank the Reviewer #1 for this comment. We have made the corrections in the revised manuscript Lines 32-33.

Minor point #2:

Line 49 – Please specify “Canada” rather than this country.

Response:

Here, “this country” was replaced by “New Zealand” (Line 49) as this is the country studied in reference [8] Farnworth MJ, Adams NJ, Keown AJ, Waran NK, Stafford KJ. Veterinary provision of analgesia for domestic cats (Felis catus) undergoing gonadectomy: a comparison of samples from New Zealand, Australia and the United Kingdom. N Z Vet J. 2014;62(3):117–22. doi: 10.1080/00480169.2013.852447

Minor point #3:

Line 76: Please describe more as to what Dispomed did as a contributor for survey distribution. It is sort of ambiguous at this point.

Response:

Dispomed database was used to send the survey electronic link to all their small animal veterinarian customers. The information was specified in revised manuscript Line 76-77.

Minor point #4:

Line 102: I think it is important to consider that your demographic data may be skewed because there is no information describing the types of practice that the survey was distributed too via dispomed. Therefore, I don’t believe it is “interesting” that majority of respondents were GPs in large or small towns because that may have been the predominant facilities in which the survey was delivered too. Perhaps just state the findings of the demographics without drawing too much conclusions from it.

Response:

We thank Reviewer #1 for this comment and the sentence has been modified in the revised manuscript, Line 102-104.

Minor point #5:

Table 1

Can you better define the term on-call hours? I think you mean 5pm-8a/weekend? Also, what would you define as episodic? Lastly, can you specify what you mean by “years of graduation”. I think you mean years since veterinary school graduation. Years of graduation doesn’t really define anything.

Response:

“On-call hours” refers to moments when practitioners are not working at the clinic but can be called for a specific emergency and have to come in to assess patients or perform emergency surgery, whether during business hours or not. A complete definition was not provided in the questionnaire. “On-call hours” was changed for “on-call duty” and the definition was added in the manuscript.

“Episodic” was used to describe any other frequency than those already mentioned in the questionnaire. To avoid confusion, “Episodic” was replaced for “Other” and specific definitions were given in Table 1.

“Year of graduation” was indeed referring to “Years of practice since veterinary school graduation”. We thank Reviewer #1 to have point this out and we have made the correction in the revised manuscript in Table 1 and modified the legends (Line 107-112).

Minor point #6:

Line 135: can you define what was considered pediatric? Since geriatric is used often we should define this. 

Response:

The provided definition was “Pediatric patients are classically under 3 months old, or as you define them in your practice”. No specific definition was provided in the questionnaire for “geriatric patients”, therefore it was indicated for the respondent “as you define them in your practice, for example 8 years for dog and 11 years old for cat”. 

Minor point #7:

Line 158: Hindsight but would have been interesting to identify the schools of graduation. 

Response:

Indeed, this would be an interesting risk factor to consider in future studies. 

Minor point #8:

Line 161 – “blood” urea “nitrogen” evaluation

Response:

We thank Reviewer #1 for this comment, and we added this precision in the revised manuscript, Line 158 and Table 4.

Minor point #9:

Table 3

What was classified or how was it determined for patients to be “believed at risk”?

“blood” urea “nitrogen and creatinine

Define the abbreviation ECG

Response:

There was no definition provided in the questionnaire for “Believed at risk”. It was voluntarily left open and for the respondent to interpret. In retrospective, only 50% (57/115) respondent used ASA status, therefore a more precise definition might have yielded a lower response rate.

We added the precision to “Blood urea nitrogen” in Table 4 as suggested by Reviewer #1.

We replaced “ECG” for “Electrocardiogram” in Table 4 as suggested by Reviewer #1.

Minor point #10:

Line 238 – this goes for all areas where only percentages are shown for comparison. Since the number of respondents that answered that specific question differ, do you think it would be of interest to your readers that you provide them with the exact count for comparison rather than or in addition to percent? I understand it gets to be a bit cumbersome, however im not sure in this situation percentage really tells us an entire lot. Just for example purposes, I think 3/10 is way different than 30/100 despite both being 30%.

Response:

We thank Reviewer #1 for this comment. We agree with this point and therefore added all counts and number of respondents corresponding to percentages cited. 

Minor point #11:

Table 4- please provide number of respondents for each item.

Response:

The number of respondents for each item was added in Table 4 (now Table 6 in revised manuscript) as recommended.

Minor point #12:

Line 273: what was the most common opioid for those graduated >15 years ago? If there was a more common one.

Response:

At Line 268-269 in Revised manuscript, hydromorphone is the most commonly used opioid for dogs by both respondents graduated over 15 years ago and less than 15 years ago, but a higher proportion of those graduated less than 15 years ago prefer it. The sentence was modified to clarify this information.

Minor point #13:

Line 279 – infusion not perfusion

Response:

The word “perfusion” was replaced by “infusion” as recommended at Line 276 in revised manuscript.

Minor point #14:

Line 282 “nerve” blocks

Response:

The word “nerve” was added at Line 280 in revised manuscript.

Minor point #15:

Line 283 “nerve” blocks

Response:

The word “nerve” was added at Line 281 in revised manuscript.

Minor point #16:

Line 284 – “infiltrative incisional line” and “ intratesticular”

Response:

We thank Reviewer #1 for this comment. The mentioned corrections were made at Line 281 in revised manuscript.

Minor point #17:

Line 285 – please make sure we use appropriate terminology “nerve block”

Response:

The word “nerve” was added at Line 283 in revised manuscript.

Minor point #18:

Table 5 – what do you mean present in the clinic? Is that number of respondents? Please provide the numbers in all percentages. Again because it depends on the number of respondents I think it would be more useful. Also for the tables I think the title should be more descriptive. It really doesn’t say much about the study. In my opinion it should be stand alone so the reader can view the table and understand the basics of the study.

Response:

“Present in the clinic” means the monitoring device is physically at the clinic and available for use by the practitioners, therefore this precision was added to the Table 8. Since the number or respondents was added for all percentages, the first column was deleted. The title of all tables were adjusted accordingly to provide more detail.

Minor point #19:

Results in general – I appreciate the work you put into providing the vast amount of detail in the results. It is fairly lengthy and I am afraid you will begin to lose some of the readers because of its length (maybe I am wrong). Perhaps with better descriptors in your tables you can shift some of this information including p values and statistical significance for comparison in tables and reduce the text length.

Response:

We thank Reviewer #1 for this comment. In an effort to reduce the length of the manuscript, some information was removed from the text when they were already presented in Tables 1 and 2. Furthermore, several other tables were added to lighten results text section.

Minor point #20:

Line 433 can we separate this sentence up into 2? Also specify younger graduates are more likely when compared to who? We assume older graduates but please specify.

Response:

As recommended, the sentence was separated, and we added that the younger graduates were compared to older graduates in Line 424-425 of revised manuscript.

Minor point #21:

Line 439 – I get what you are saying but I do not think I would write that in an open access journal based on it being a big speculation.

Response:

We thank Reviewer #1 for this comment. We understand that since this is a speculation it is not pertinent in an open access journal and the sentence was removed from revised manuscript (Line 428).

Minor point #22:

Line 467 – I don’t agree with this statement. Yes, it does cause increased resistance but not if positive pressure ventilation is provided (by hand or machine). In addition the studies used to assess resistance are fairly old and used metal valves which were heavy. Our university does not use NRB circuits just because of the concern a student will accidently hit the quick flush. 

Response:

Even though it is possible to anaesthetise various patient sizes with only one type of anaesthetic system, it might be suboptimal to use Bain circuit in large patients or rebreathing system in small patients for the reasons mentioned in the manuscript. We understand Reviewer #1’s comment and we modified the paragraph starting Line 457-460 in revised manuscript to nuance our statements.

Minor point #23:

Line 475 – I think simon et al 2017 on oligoanalgesia would be a good reference for this sentence.

Response:

We thank Reviewer #1 for this comment. It is indeed a pertinent reference discussing the consequences of oligoanalgesia. Therefore, it was added to sentence in Line 465 of revised manuscript as Reference 21.

Minor point #24:

Line 480-481 – I think you need to support this with a reference. Many areas routinely give them postoperatively so perhaps a brief explanation as to why this is inappropriate in Canada.’

Response:

These sentences were modified to better express our thoughts in Line 469-470 of revised manuscript.

Minor point #25:

Line 490 – lets avoid restating the results in the discussion please.

Response:

The discussion was revised entirely to limit results repetition.

Minor point #26:

Line 502 – inexpensive

Response:

This orthographic error was corrected in Line 489 of revised manuscript.

Minor point #27:

line 508 – to administer “fluid”? be more specific

Response:

We thank Reviewer #1 for this comment. We specified “fluid therapy” instead of “fluid” in Line 495 of revised manuscript.

Minor point #28:

line 516- thanks for pointing this out

Response:

We thank Reviewer #1 for this comment. The authors felt indeed the need to bring this precision, so the reader doesn’t misinterpret the risks related to endotracheal intubation in cats for a recommendation against it.

Minor point #29:

Line 521 – again too much results in the discussion. I understand there has to be some reference but in an already long manuscript I would try to reduce this when possible.

Response:

To help decrease result repetition, several sentences were deleted from revised manuscript.

Minor point #30:

Line 535 – who’s recommendations?

Response:

It is American Animal Hospital Association’s recommendation. It is now specified in Line 518 of revised manuscript.

Minor point #30:

line 602- please state a sig proportion of those surveyed… again I don’t think just over 100 vets is a sig portion of the population in Canada.

Response:

The word “significant” was removed and it was specified that it’s the surveyed population that don’t respond to guidelines, as recommended in Line 584 of revised manuscript.

Editor

Dear Dr. Troncy,

Thank you for submitting your manuscript to PLOS ONE. After careful consideration, we feel that it has merit but does not fully meet PLOS ONE’s publication criteria as it currently stands. Therefore, we invite you to submit a revised version of the manuscript that addresses the points raised during the review process.

Response:

We thank the Editor about his general comment concerning our manuscript. As recommended by the Reviewer, the modifications have been carried out and the essential revisions have been clarified in the answers addressed for each point. Thank you.

---

## [Decision Letter · Decision Letter 1]

12 Aug 2021

PONE-D-21-05375R1

Management of veterinary anaesthesia and analgesia in small animals: A survey of English-speaking practitioners in Canada

PLOS ONE

Dear Dr. Troncy,

Thank you for submitting your manuscript to PLOS ONE. After careful consideration, we feel that it has merit but does not fully meet PLOS ONE’s publication criteria as it currently stands. Therefore, we invite you to submit a revised version of the manuscript that addresses the points raised during the review process.

The reviewer has still concerns regarding the statistical analysis in some areas (comparing small vs. larger groups). Please address these concerns in your discussion. Furthermore, the reviewer also noticed that there are many grammatical errors throughout your manuscript.

We look forward to receiving your revised manuscript.

Kind regards,

Jan S Suchodolski, DVM, PhD

Academic Editor

PLOS ONE

Journal Requirements:

Reviewers' comments:

Reviewer's Responses to Questions

**Comments to the Author**

1. If the authors have adequately addressed your comments raised in a previous round of review and you feel that this manuscript is now acceptable for publication, you may indicate that here to bypass the “Comments to the Author” section, enter your conflict of interest statement in the “Confidential to Editor” section, and submit your "Accept" recommendation.

Reviewer #1: All comments have been addressed

2. Is the manuscript technically sound, and do the data support the conclusions?

Reviewer #1: Yes

3. Has the statistical analysis been performed appropriately and rigorously? 

Reviewer #1: No

4. Have the authors made all data underlying the findings in their manuscript fully available?

Reviewer #1: Yes

5. Is the manuscript presented in an intelligible fashion and written in standard English?

Reviewer #1: Yes

6. Review Comments to the Author

Reviewer #1: Thank you so much for addressing all of my concerns. The manuscript is much approved. There are still some grammatical errors throughout that require the editors attention. I just had a few more minor comments

Line 20: Perhaps it is a common term in Canada (I will leave it up to the authors), but “English Canada” seems like an awkward term to me. Makes it sound like it’s a territory in Canada operated by the English.

Line 21: SurveyMonkey® versus SurveyMonkey?

Line 27: 126/? Respondents.

Line 75,87: SurveyMonkey® ?

Line 107: I guess I would consider these vets still working at the clinic, they just are not present at the time of the animal’s presentation.

Line 202: I think this is a confusing statement “(81% (38/47) use it more than 20% of times vs. 61% (30/49),”. What is confusing is “20% of times” do you mean “20% of the time”?

Line 204: same for here ”(22% (16/72) use it in more than 20% cases vs. 67% (6/9), P=0.023”. 20% of cases.

Line 225: This is why I asked about total participants “Respondents working in referral centres use ketamine-medetomidine (75% (6/8) vs. 29% 226 (16/56), P=0.016)”. I don’t understand how you found this to be statistically significant with only 8 individuals in 1 group. Population of 8 vs 56 doesn’t seem appropriate for comparison.

Line 226: same as line 225.

Line 235: Again im not sure comparing such a small sample size to a large sample size is appropriate.

7. PLOS authors have the option to publish the peer review history of their article (what does this mean?). If published, this will include your full peer review and any attached files.

Reviewer #1: No

---

## [Editor Report · Decision Letter 2]

2 Sep 2021

Management of veterinary anaesthesia and analgesia in small animals: A survey of English-speaking practitioners in Canada

PONE-D-21-05375R2

Dear Dr. Troncy,

We’re pleased to inform you that your manuscript has been judged scientifically suitable for publication and will be formally accepted for publication once it meets all outstanding technical requirements.

Kind regards,

Jan S Suchodolski, DVM, PhD

Academic Editor

PLOS ONE
---

## [Editor Report · Acceptance letter]

20 Sep 2021

PONE-D-21-05375R2 

Management of veterinary anaesthesia and analgesia in small animals: A survey of English-speaking practitioners in Canada 

Dear Dr. Troncy:

I'm pleased to inform you that your manuscript has been deemed suitable for publication in PLOS ONE. Congratulations! Your manuscript is now with our production department. 

Kind regards, 

on behalf of

Dr. Jan S Suchodolski 

Academic Editor

PLOS ONE